# Adjustment of Tall Building Behavior by Guided Optimization of Magneto-Rheological Damper Control Parameters

Amin Akhnoukh [1,*], Ahmed Fady Farid [2], Ahmed M. M. Hasan [2,3] and Youssef F. Rashed [2]

1 Construction Management Department, East Carolina University, Greenville, NC 27858, USA
2 Structural Engineering Department, Cairo University, Cairo 12613, Egypt
3 Civil and Infrastructure Engineering and Management (CIEM) Program, School of Engineering and Applied Science, Nile University, Juhayna Square, 26th of July Corridor, Giza 12588, Egypt
* Correspondence: akhnoukha17@ecu.edu; Tel.: +1-501-249-7961

**Abstract:** Magneto-rheological dampers (MR-Dampers) are increasingly being used in construction applications to reduce the dynamic response of structures to seismic activities or severe wind loading. Sensors attached to the structure will signal the computer to supply the dampers with an electric charge that transfers the MR fluid to a near-solid material with different physical and mechanical properties (viscoelastic behavior). Control algorithms govern the fluid to near-solid conversion, which controls the behavior of the damper and the performance of the structure under the seismic or wind loading event. The successful optimization of control parameters minimizes the overall structural response to dynamic forces. The main objective of this research is to change the output behavior of specific floors within a building subjected to seismic excitation by optimizing the MR-Damper control parameters to impact the behavior of a specific floor or number of floors within the building. The adjustment of control parameters to attain this objective was validated in multiple case studies throughout this research. The successful implementation of the research outcome will result in optimized MR-damper design to meet the performance-based criteria of building projects.

**Keywords:** MR-Dampers; structural control; MR fluids; seismic excitation; smart buildings





## 1. Introduction

Building structures and bridges are vulnerable to shocks due to seismic loading. In recent years, major earthquake events resulted in the death of thousands and in billions of dollars of asset losses. Solutions to mitigate earthquake losses include the development of performance-based design codes, the use of high-performance construction materials with improved durability and long-term performance [1–5], and self-healing materials. One of the recent technologies introduced is fitting damping devices into structural members that are prone to seismic activities. There are different types of damping devices with different characteristics and functionality, including (1) friction dampers (FDs), tuned mass dampers (TMDs), viscous dampers (VDs), and magneto-rheological dampers (MRs). The aforementioned types of dampers increase the safety of buildings by dissipating the energy of seismic excitation. Thus, they minimize the seismic-induced stresses in any building. The functionality of these dampers is explained as follows:

Friction dampers (FDs): FDs are designed to slip before the building structural integrity is affected. Seismic energy buildup is generated within friction dampers until it overcomes the damper frictional resistance. Hence, damper surfaces slide against each other and release heat to dissipate the energy. Friction dampers do not require replacement posts for any seismic activity [6–9].

Tuned mass dampers (TMDs): Also known as harmonic absorbers, these devices are mounted in structures to reduce lateral vibrations due to seismic activities. The TMD consists of a mass mounted on one or more damped springs. The oscillation frequency

of the TMD is similar to the resonant frequency of the structure it is mounted to, and the reduces the structure's maximum vibration under earthquake activities [10–12].

Viscous dampers (VDs): Also known as seismic dampers, these are hydraulic devices used to dissipate the kinetic energy induced by earthquakes. VDs allow for free and controlled damping of structures. VDs are designed and installed such that their behavior in damping vibrations depends on the excitation of the structure. Thus, VD behavior varies when damping earthquakes or during wind loading [13–16].

Magneto-rheological dampers (MRs): Magneto-rheological dampers (MR-Dampers) are increasingly used in construction applications to reduce the dynamic response of structures to seismic activities or severe wind loading. Sensors attached to the structure will signal the computer to supply the dampers with an electric charge that transfers the MR fluid to a near-solid material with different physical and mechanical properties (viscoelastic behavior) [17–21].

The main objective of this research is to control the response behavior of specific floor(s) in a building to meet specific performance criteria under earthquake excitation using MR-Dampers. The research objective was attained by optimizing the MR-Damper control parameters to attain the required response behavior. The adjustment of control parameters to attain this objective was validated in multiple case studies throughout this research. The adjustment of MR-damper control parameters to target a given performance of specific floors resulted in the optimized design of MR-damper-installed building structures.

## 2. Literature Review

MR-Dampers are increasingly used in the seismic design of building structures, civil infrastructure [22,23], semi-active vibration isolation systems [24–26], and military vehicles [27]. MR-Dampers are characterized by low energy consumption for high damping forces and for their fast response. MR dampers depend mainly on MR fluids, which are considered intelligent fluids due to their ability to change their properties when exposed to a magnetic field. MR fluid's main constituents include the carrier fluid, magnetic particles, and additives. Carbonyl iron, with 99% purity, is often used as magnetic particles due to its high magnetic permeability [28]. In some MR-Dampers, carbonyl iron particle sizes range between 3 to 5 microns and have a concentration ranging from 20% to 40%. Thus, the likelihood of corrosion of the MR-Damper outer surface is minimal [29]. When a magnetic field is applied to the MR fluid, the fluid rheological properties change, and the shear capacity of the fluid increases. The magnetic field is applied by a wire coil embedded within the damper [30]. A detailed diagram of MR-damper function is shown in Figure 1, and an MR-Damper structure is shown in Figure 2.

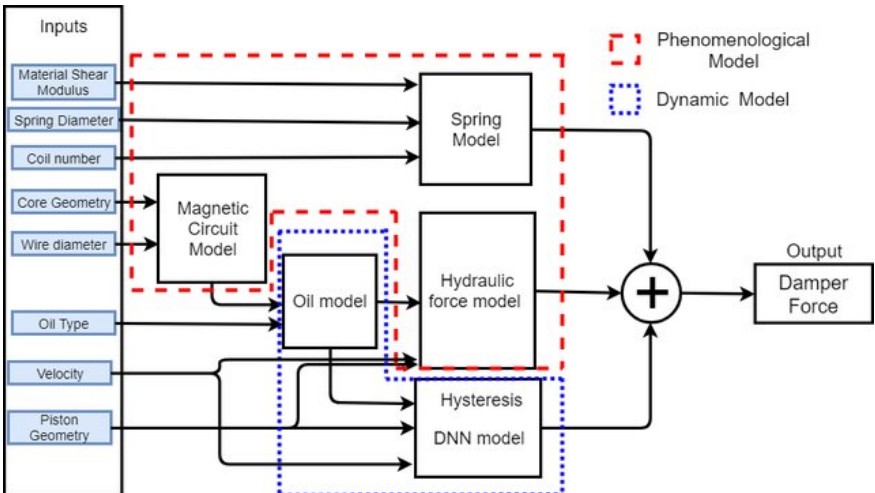

**Figure 1.** Schematic diagram of MR-Damper semi-active control system. Reprinted from Ref. [31].

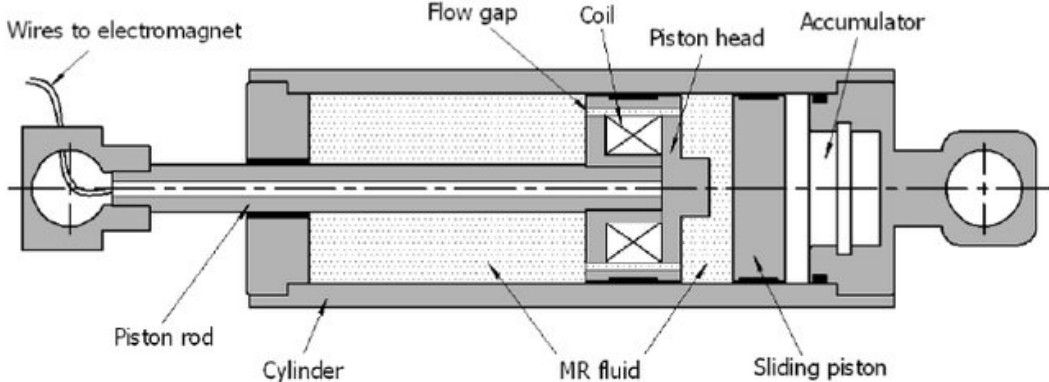

**Figure 2.** MR-Damper structure [32].

Researchers investigated different strategies to control MR-Dampers. In a relevant research study, researchers proposed using a clipped optimal (CO) controller and a Lyapunov controller control. The study investigated the behavior and response of a six-story test structure with four shear mode MR-Dampers; two dampers were placed between the base and the first floor, and two dampers were placed between the first and the second floor. The results obtained from experimental data were comparable to an equivalent passive control system of control MR-Dampers to reduce structural responses due to seismic loads.

In 2002, Yang et al. presented the dynamic model of the damper [33]; later, in 2004, they extended the dynamic model to include the phenomenological model of the damper that demonstrated the MR-Damper potential for practical civil engineering applications [34]. In 2003, Fujitani et al. [35] developed a 400 kN bypass-type damper for base isolated buildings using a new MR fluid. Experimental testing validated the dynamic model and the capacity of the developed damper. Later, in 2013, a MR-Damper design with numerical example for shear-valve mode was carried out by Xu, et al. [36], where they designed and tested a 200 kN damper. Onoda and Oh et al. [12,13] demonstrated the superiority of MR-Dampers (flow bypass type) over ER dampers at suppressing the vibration of a 10-bay truss system.

Control algorithms play a key role for structurally controlled buildings. In a comparative study conducted in 2000, Jansen and Dyke [37] presented the results of a series of control algorithms applied to MR-Dampers for a six-story numerical model. Later, more advanced control algorithms based on fuzzy controllers emerged. These algorithms can produce optimized performance, such as the ones presented by Choi, et al. [15] in 2004 and by Bhardwaj and Datta [38] in 2006. By 2010, a direct adaptive controller was presented by Bitaraf, et al. [39]. In 2011, a genetic based fuzzy controller was presented by Cetin et al. [18]. In 2013, Ali [40] presented a Quasi-Bang-Bang controller with fuzzy-logic-based input for the driver voltage, and, in 2019, Bozorgvar and Zahrai [41] presented an adaptive neural-fuzzy intelligent controller that was optimized using a genetic algorithm.

In 2018, Bathaei et al. [42] presented an 11-DOF building with a toned mass damper, two MR-Dampers, and a fuzzy control algorithm to evaluate their capacity in seismic mitigation compared to an uncontrolled building. Later, in 2021, Bagherkhani and Baghlani [43] presented a reliability assessment of MR-Dampers for two cases: structural safety and human comfort when used in the semi active and passive modes. Also, Fakhry et al. [44] presented an optimized response for a practical high-rise structure that was modeled using a mixed finite element method and a boundary element method (BEM) for nine practical buildings with 20 floors each. The authors in [44] used a control algorithm based on the clip optimal approach and variable voltage for the current driver approach.

In this research, the previous methodology developed by Fakhry et al. [44] was implemented using a graphic user interface (GUI). The GUI exploits the full potential to guide the control parameters for an optimized response. This guidance is performed to satisfy a specific floor or set of floor design criteria. The new setup was evaluated for potential

favorable control output using practical buildings and earthquakes that were both synthetic and historical.

In order to facilitate current testing and future utilization, the GUI was also split into three modes, which are described as follows:

Design mode: This uses one earthquake to generate control parameters.
Simulation mode: This is used to produce a controlled response for new earthquakes.
Acting mode: This is a generated code that is installed on a special purpose computer or a micro controller to apply voltage to MR-Dampers in situ.

### 3. Graphical User Interface (GUI) Smart Control

To facilitate the validation of the proposed scheme, several steps were defined. First, an interface for the smart controller was developed. The interface was split into two tabs.

The first tab (design mode) contained all required data input parameters for the design of the MR-Damper placement, as presented in Figure 3. After the building properties directory is selected, and all required structural data is loaded in the first GUI container (1. Building Properties Directory), an earthquake for design mode is either selected from earthquakes used in a previous dynamic analysis of a building or the user can select a new design earthquake in the second GUI container (2. Loading Design EQ).

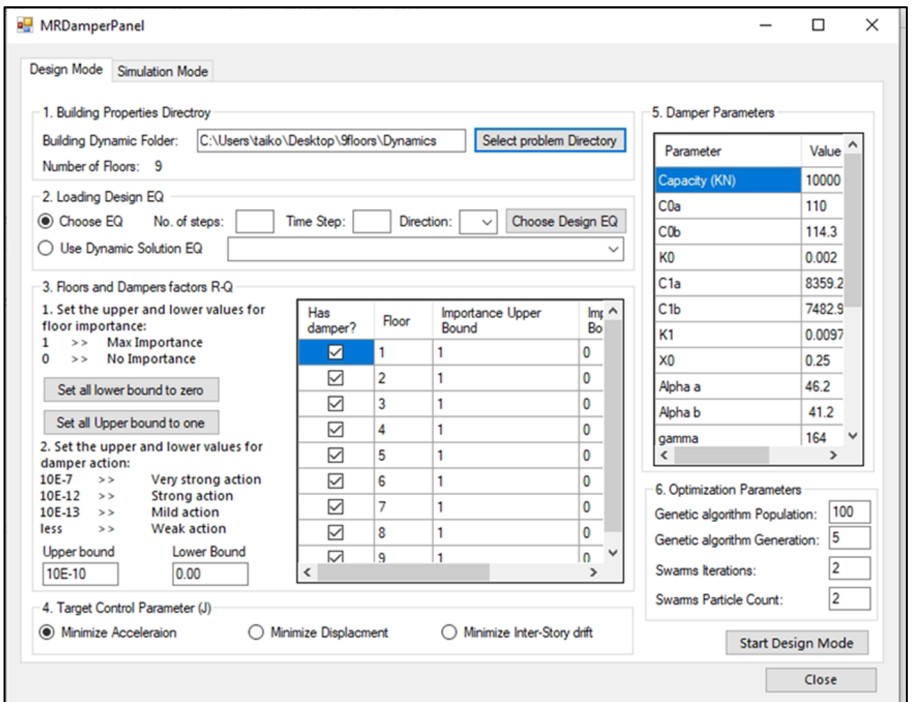

**Figure 3.** Design mode graphical user interface (GUI).

In the third container of the GUI (3. Floors and Damper Factors R-Q), the user defines the floors with potential damper placement and defines the upper and lower bounds for the R control parameter that is passed to the PSO according to required design criteria; also, the upper and lower bounds for the Q control parameter are passed to the PSO.

In the fourth container (4. Target Control Parameter J), the target response minimization type is selected from three evaluation criteria: Displacement, Acceleration, and Inter-story drift. These three options correspond to $J_1$, $J_2$, and $J_3$, respectively, which were defined in previous research [45]. The GA target objective function changes according to user input. In the fifth container (Damper Parameters), the user inputs the MR-Damper parameters according to the modified Boch–Wen model presented in Appendix B.

The last container (5. Optimization Parameters) contains the desired GA optimization population and generations count, as well as the PSO number of iterations and the particle count.

After filling all required data, the user starts the design mode, which is presented in detail in Section 3 (Control Scheme).

The second tab (Simulation Mode) presented in Figure 4 collects the available earthquakes from previous dynamic analyses. In addition, it allows the user to add new earthquakes for simulation. After selecting one or several Earthquake files, the controlled response of the building for the selected earthquake is generated using previously produced control gain parameters and generated damper locations. The simulation mode operation is presented in the following section.

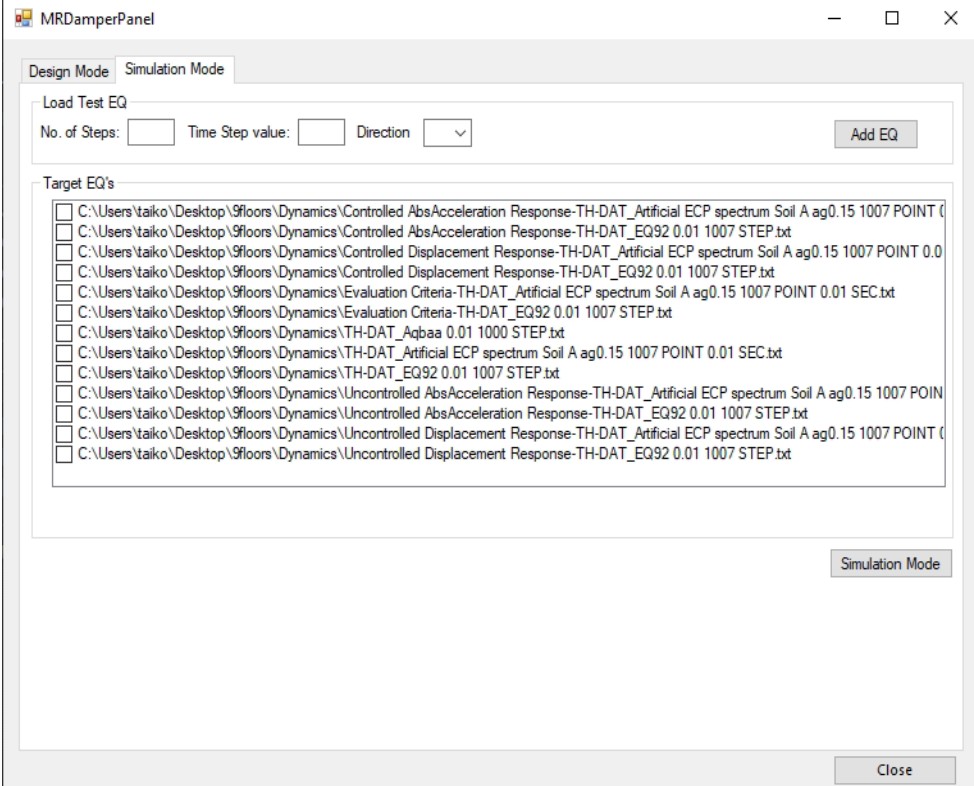

**Figure 4.** Simulation mode graphical user interface (the data of the available earthquakes).

## 4. The Control Scheme

Fakhry et al. [37] presented the force-ratio-based voltage control scheme that was implemented for this study; in this approach, the voltage for the MR force is calculated based on the ratio between the current force in the damper ($f$) and the desired control force ($f_c$), as presented in Equation (1). The graduation of voltage change proved to be more efficient compared to the CO approach.

$$V = \begin{cases} \left(1 - \frac{f}{f_c}\right) V_{max} & f < f_c \\ 0, & f > f_c \end{cases} \tag{1}$$

The operation scheme is split into three modes of operation. The first mode is as described in the previous section, where the software controls the response of a building for a given earthquake, and the output of this mode is a controlled building response with dampers locations and control gain weighting matrices (R and Q) (see Appendix A).

The second mode is a simulation mode, which generates a new earthquake along with previously generated damper locations and control gain weighting matrices (R and Q)

by using previously generated building data. A controlled response is generated for the new earthquake.

The third mode is an embedded system operation. This mode is like the simulation mode, but all variables for the software (i.e., building stiffness matrix, mass matrix, damping matrix, control gain parameters, damper locations, and electric current ports) are static and imbedded in the system, except for sensor data, which is obtained directly from accelerometer sensors located at each floor. The earthquake record is obtained from the national seismic network or directly from a nearby seismological recoding station.

### 4.1. Design Mode

Figure 5 presents the overall control flow of the implemented software. The software begins by obtaining building data that results from the boundary element method and finite element method (BEM/FEM) analysis performed in a BEM environment [46]. The earthquake data is also loaded into the same environment.

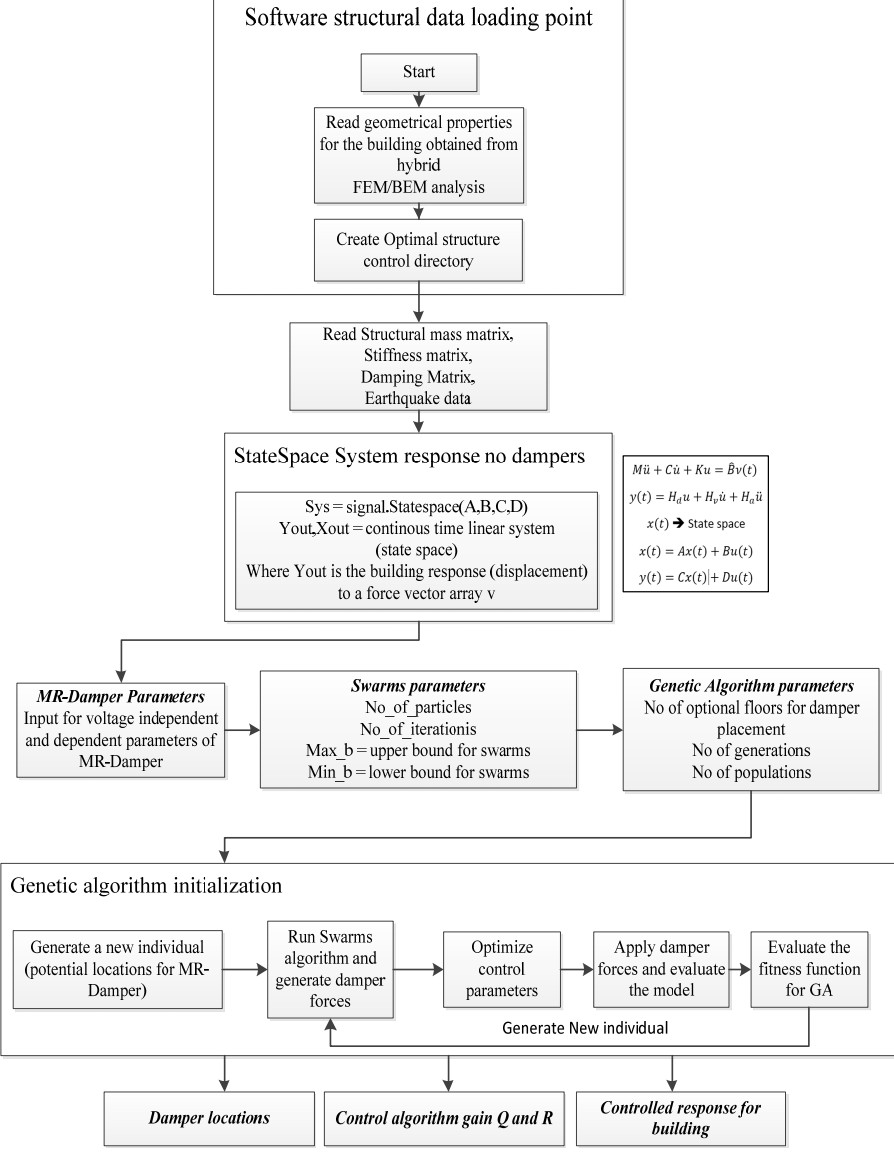

**Figure 5.** Flow chart for the control scheme (design mode).

The state-space representation of the building is generated without control forces. The next step for the software is to obtain the values for the parameters of the MR-Damper, the

parameter for the genetic algorithm, and the parameters for the particle swarm algorithm (PSO) [47].

The main program loop is the genetic algorithm. The genetic algorithm generates an individual element, and this individual element is a set of trial locations for the MR-Damper that feeds the damper location to the second loop of the software to the PSO. The PSO optimizes the Linear Quadratic Gaussian (LQG) control algorithm by changing the weighting matrices Q and R. The PSO runs the control algorithm and estimates control gain parameters Q and R in the process to minimize a fitness function J. (See Appendix A). These control parameters are stored for each iteration of the PSO. Finally, after finding the best operation control gain and best damper locations, the best damper locations are stored, and the controlled response of the building is generated and stored.

*4.2. Simulation Mode*

The simulation mode is presented in Figure 6. The resulting control gain, damper location, and building parameters are combined with a new earthquake and produce the corresponding controlled response.

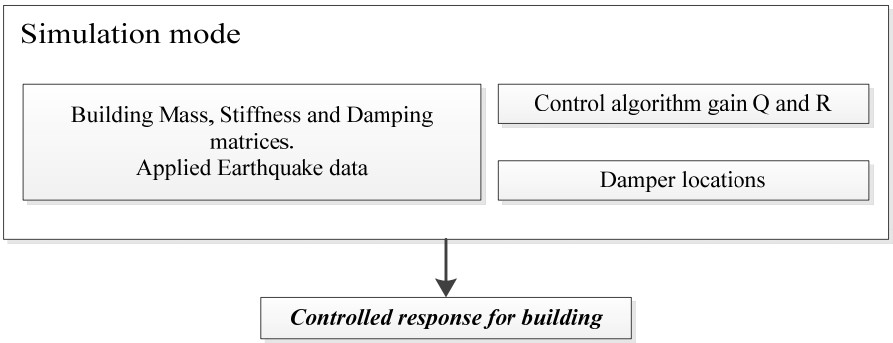

**Figure 6.** Simulation mode.

*4.3. Acting Mode*

The acting mode presented in Figure 7 is a fixed code to be installed on a special purpose computer (SPC) or a digital controller. The building properties, including the Stiffness matrix, Mass matrix, and Damping matrix, are stored in fixed format in the SPC. Also, the weighting matrices R and Q are generated from the design mode after sufficient iteration for the double optimization scheme. Sensors to be installed in the building, including the number of sensors, are obtained from the GA. When the structure is excited from an earthquake, the current driver provides voltage to the MR-Dampers that matches the resulting voltage required to mitigate the building response according to the building response type, which is also stored on the SPC.

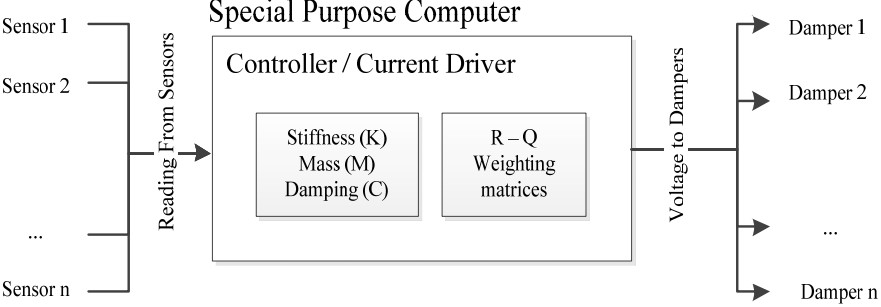

**Figure 7.** The schematic for the operation of the acting mode.

**5. Case Study**

For the study of changing the control gain parameters and changing the target response type for minimizing, two buildings were selected. Figure 8 presents the typical plan of the

buildings. The building is 40 times 32 m with 49 columns, and has two three-sided cores, as well as three openings.

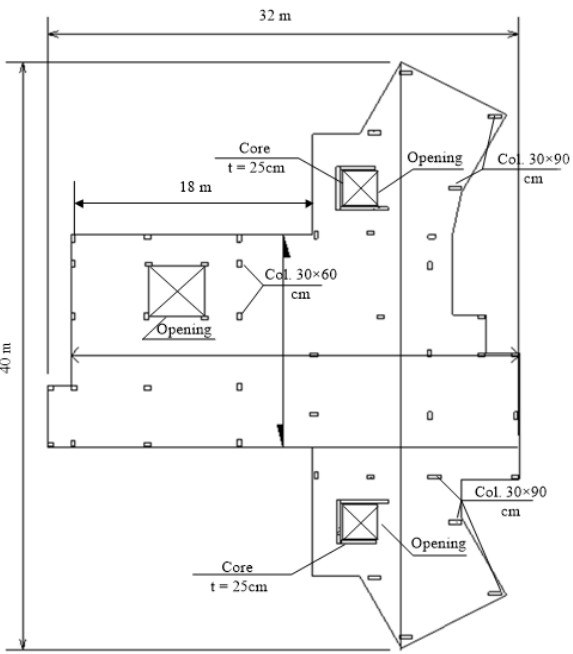

**Figure 8.** Typical floor plan for the study buildings.

Figure 9 presents the 3D view of the buildings (18 floors and 9 floors) with a typical floor height of 3 m.

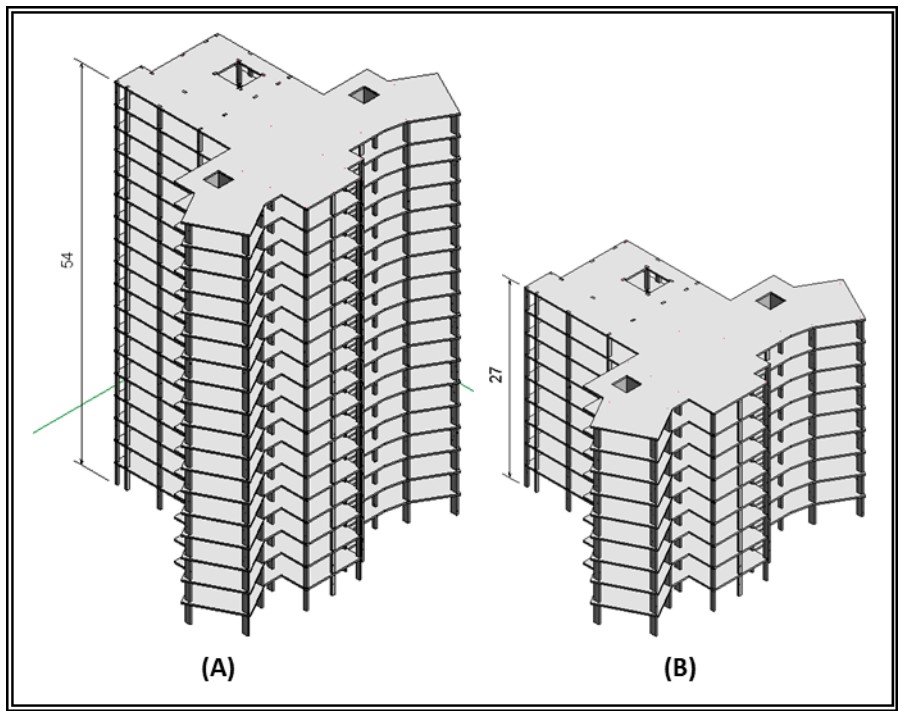

**Figure 9.** (**A**) 18-floor building and (**B**) 9-floor building.

Earthquakes were selected in two different locations for the investigation: Egypt and the USA. The design response spectrums for both locations were used to generate synthetic earthquakes. All acceleration values are expressed in meter/sec$^2$ on the Y axis and in seconds in X axis. The Egyptian earthquake was generated with $a_g$ = 0.15 g (base

ground acceleration) and Soil Type A. This earthquake is plotted in Figure 10. The ASCE earthquake generated for Soil Type C and $S_s$ = 0.6 and $S_1$ = 0.32 is plotted in Figure 11. Historical earthquakes at El Centro in 1940, Northridge in 1994, Egypt in 1992, and Aqaba in 1995 are presented in Figures 12–15 in terms of acceleration in m/sec$^2$.

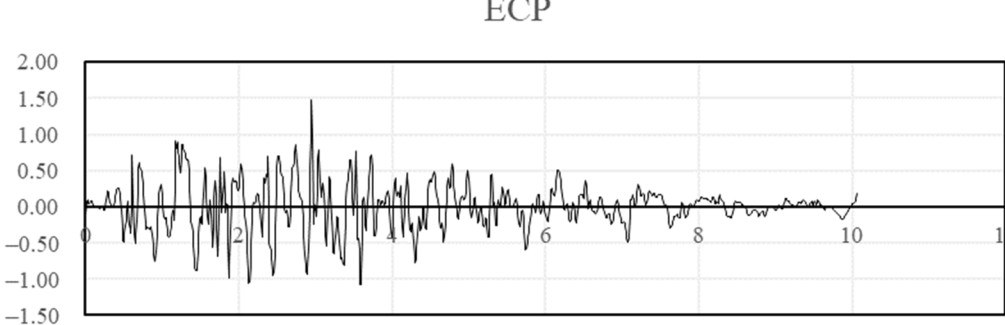

**Figure 10.** Synthetic earthquake generated from ECP.

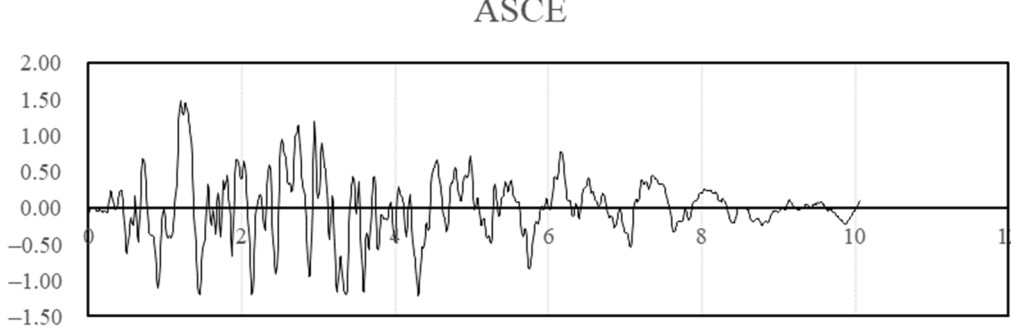

**Figure 11.** Synthetic earthquake generated from ASCE.

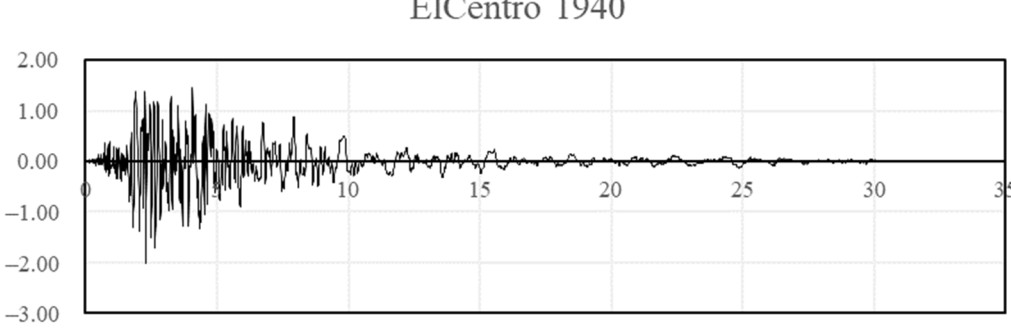

**Figure 12.** El Centro 1940 earthquake (ASCE 7–10).

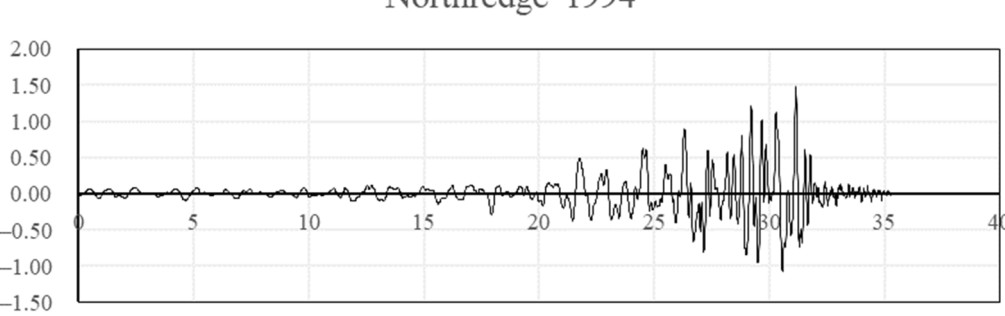

**Figure 13.** Northridge 1994 earthquake (ASCE 7–10).

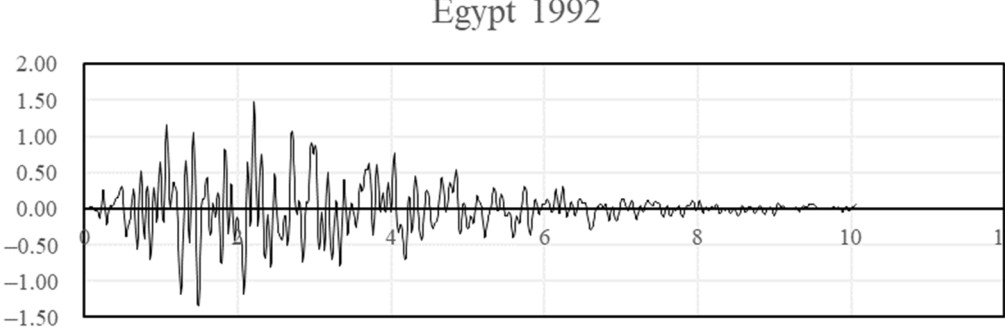

**Figure 14.** Egypt 1992 earthquake.

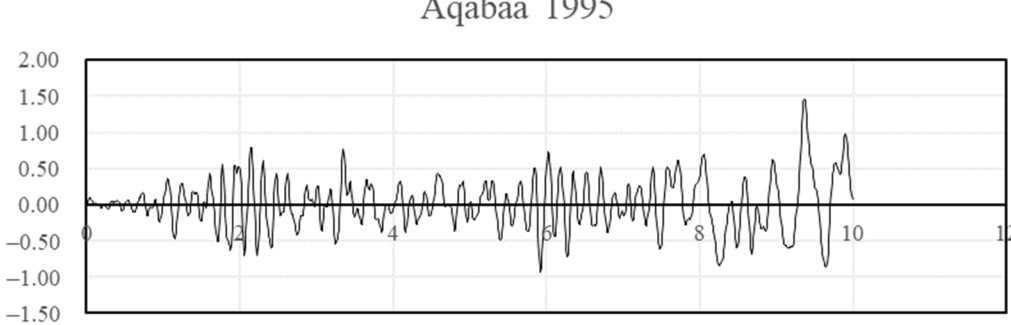

**Figure 15.** Aqaba 1995 earthquake.

The GUI allows for three types of target response optimization: $J_1$, $J_2$, and $J_3$. Switching between the three values changes the target optimization function for the GA. Twelve tests were carried out for the two buildings. The 12 design runs were preformed using the generated artificial response spectrum earthquakes. For each building, one spectrum using the ECP earthquake and one spectrum using the ASCE earthquake was applied, where the target response optimization was set to each of the three target responses. A total of 24 simulation mode models were carried out for each building in each of the two locations. Two local earthquakes were simulated for each of the three responses. Table 1 presents a summary of the analysis of the GA approach, where the letter D donates design mode, and the letter S donates simulation mode.

**Table 1.** GA target Response optimization change scheme.

| Target Response | Building 1 (9 Floors) | | | Building 2 (18 Floors) | | |
|---|---|---|---|---|---|---|
| | Acc. | Disp. | Inter-Story Drift | Acc. | Disp. | Inter-Story Drift |
| ECP | D | D | D | D | D | D |
| ASCE | D | D | D | D | D | D |
| Egypt 92 | S | S | S | S | S | S |
| Aqabaa | S | S | S | S | S | S |
| Northridge | S | S | S | S | S | S |
| El Centro | S | S | S | S | S | S |

*PSO Floor-Specific Control Weighting Parameters Change*

The GUI allows for the change of PSO ranges regarding the R and Q weighting matrices. The R weighting matrix is directly linked to the controlling force i.e., the MR-Damper action force. The PSO algorithm is allowed to change the R value freely. This means that there are no constraints applied to the damper force, since the maximum force in the damper is already limited to the damper model (presented in Appendix B).

For the Q weighting matrix, the PSO input range for each floor of the building is set to the range (0,1) by default. The PSO range is changed to (0.9,1) when the significance of a specific floor of the building needs to be increased to the overall structural behavior in terms of displacement. This practice was carried out for an array of possible scenarios. This scheme was performed four times: once for each of the two buildings and once for each of the artificial earthquakes.

Each case had one design mode using the artificial earthquake and two simulation modes using the provided earthquakes for each design code of the practice region. Selected floors for the Q modification scenarios are shown in Table 2.

**Table 2.** The selected floors for the Q modification scenarios.

| | Building 1 (9 Floors) | Building 2 (18 Floors) |
|---|---|---|
| **Case** | **Target Floors** | **Target Floors** |
| 1 | 3 | 6 |
| 2 | 6 | 12 |
| 3 | 9 | 18 |
| 4 | 3, 6 | 6, 12 |
| 5 | 3, 6, 9 | 12, 18 |
| 6 | 6, 9 | 6, 12, 18 |
| 7 | 3, 4 | 6, 7 |
| 8 | 3, 4, 5 | 6, 7, 8 |
| 9 | 6, 7 | 12, 13 |
| 10 | 6, 7, 8 | 12, 13, 14 |
| 11 | 2, 3, 4, 5 | 4, 5, 6, 7 |
| 12 | 6, 7, 8, 9 | 12, 13, 14, 15 |

## 6. Selected Results and Discussions

Figures 16 and 17 present the base case for forces in the damper when the 9-story building was subjected to the ECP EQ and ASCE EQ, respectively. The reduction in the displacement response was at 52% for the ECP EQ and 50% for the ASCE EQ. The second case had more maximum displacement and, consequently, more damping force.

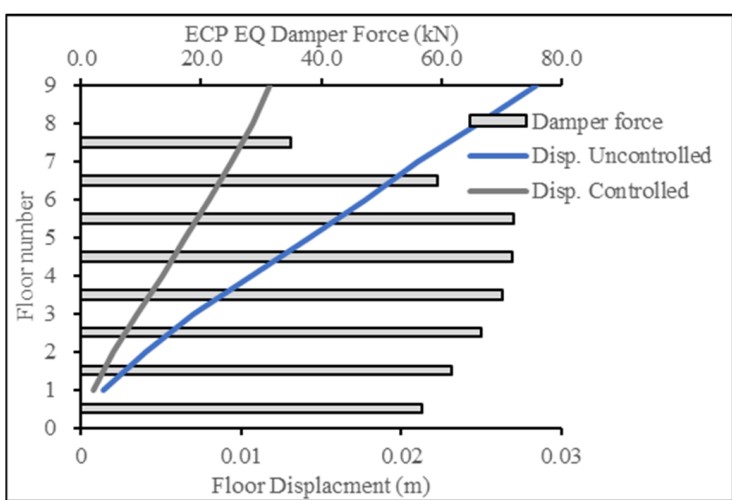

**Figure 16.** Displacement reduction for ECP EQ for 9-story building.

For Figures 18 and 19, the target response type was changed to acceleration and compared to the acceleration resulting from the base case (displacement case). The results showcase the success in producing a lower acceleration response, thereby achieving the targeted results.

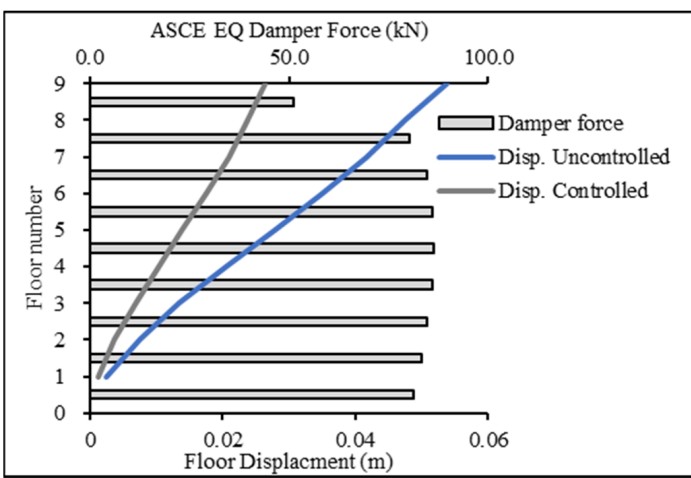

**Figure 17.** Displacement reduction for ASCE EQ for 9-story building.

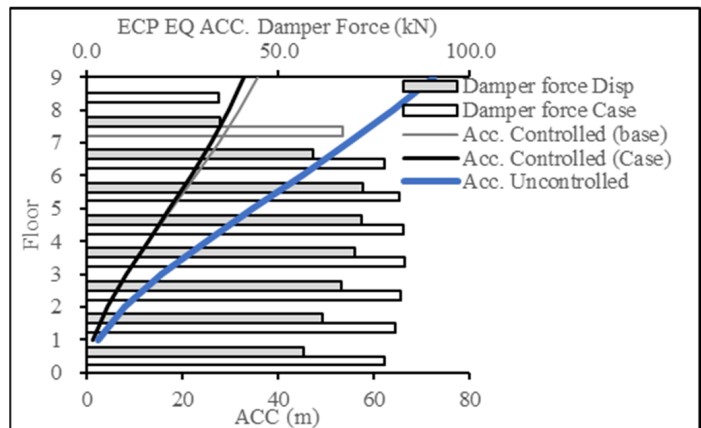

**Figure 18.** Acceleration reduction for ECP EQ for 9-story building.

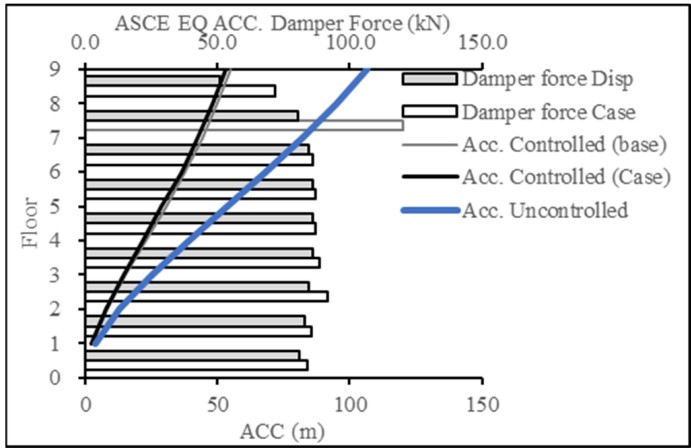

**Figure 19.** Acceleration reduction for ASCE EQ for 9-story building.

For Figures 20 and 21, the target response type was changed to inter-story drift and compared to the inter-story drift resulting from the base case (displacement case). The results showcase the success in producing a lower inter-story drift response, thereby achieving the targeted results.

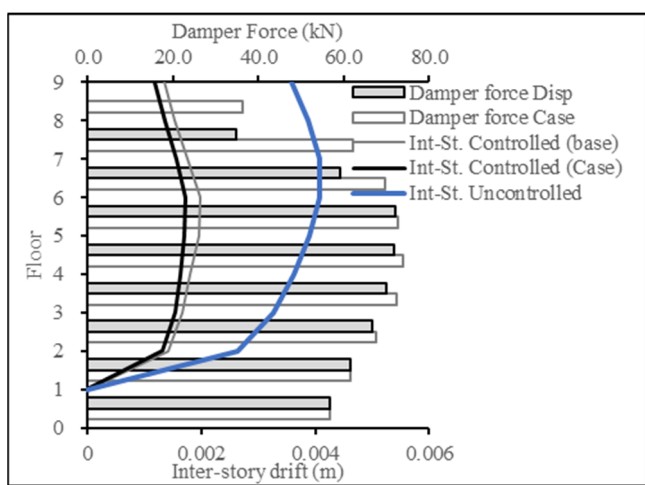

**Figure 20.** Inter-story drift reduction for ECP EQ for 9-story building.

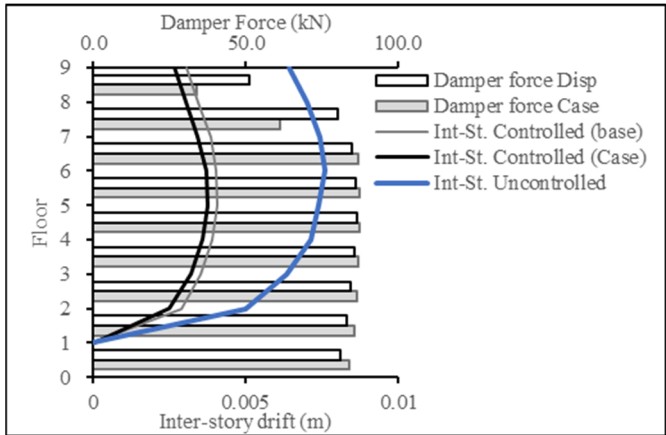

**Figure 21.** Inter-story drift reduction for ASCE EQ for 9-story building.

Figures 22 and 23 present the base case for forces in the damper when the 18-story building was subjected to the ECP EQ and ASCE EQ, respectively. The reduction in the displacement response was at 54% for the ECP EQ and 42% for the ASCE EQ. The second case had a higher maximum displacement and, consequently, more damping force.

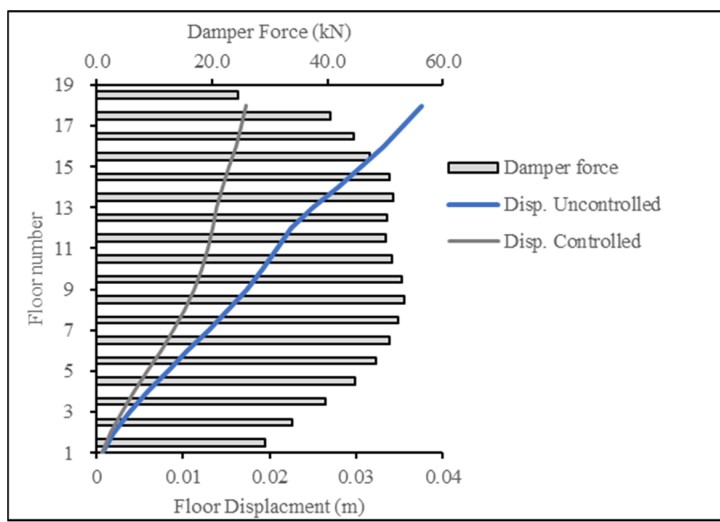

**Figure 22.** Displacement reduction for ECP EQ for 18-story building.

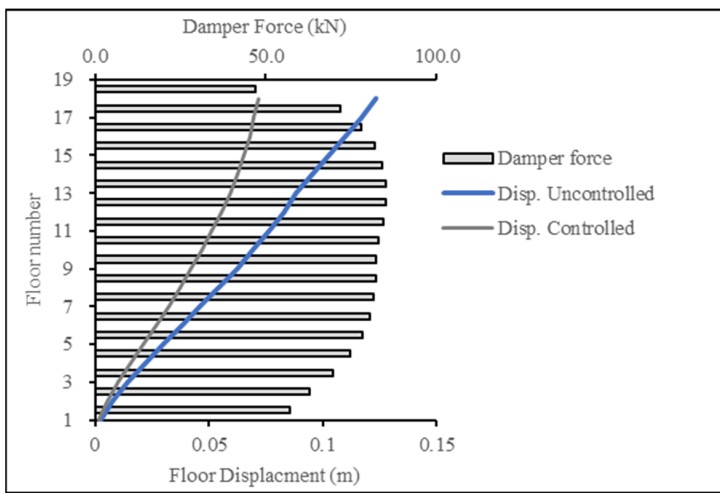

**Figure 23.** Displacement reduction for ASCE EQ for 18-story building.

For Figures 24 and 25, the target response type was changed to acceleration and compared to the acceleration resulting from the base (displacement case). The results showcase the success in producing a lower acceleration response, thereby achieving the targeted results.

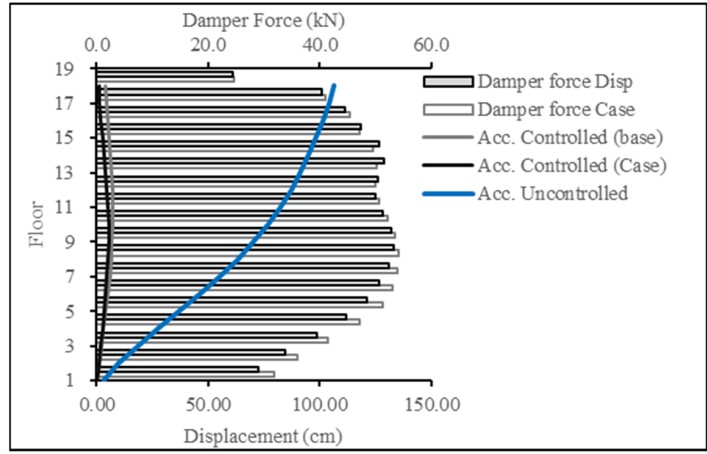

**Figure 24.** Acceleration reduction for ECP EQ for 18-story building.

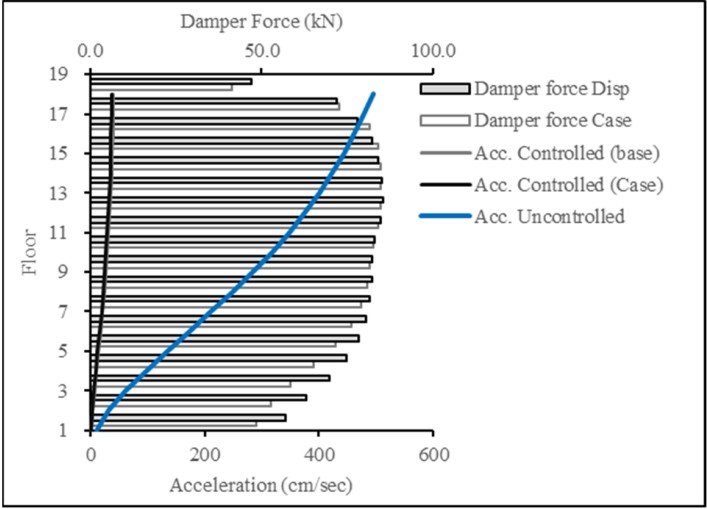

**Figure 25.** Acceleration reduction for ASCE EQ for 18-story building.

For Figures 26 and 27, the target response type was changed to inter-story drift and compared to the inter-story drift resulting from the base case (displacement case). The results showcase the success in producing a lower inter-story drift response, thereby achieving the targeted results.

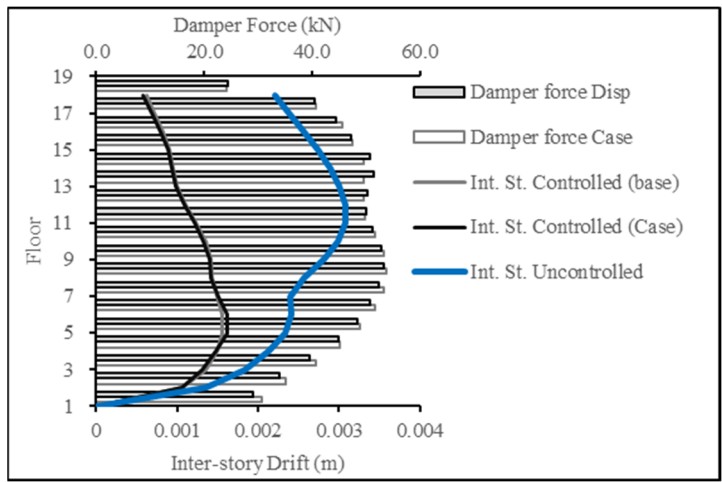

**Figure 26.** Inter-story drift reduction for ECP EQ for 18-story building.

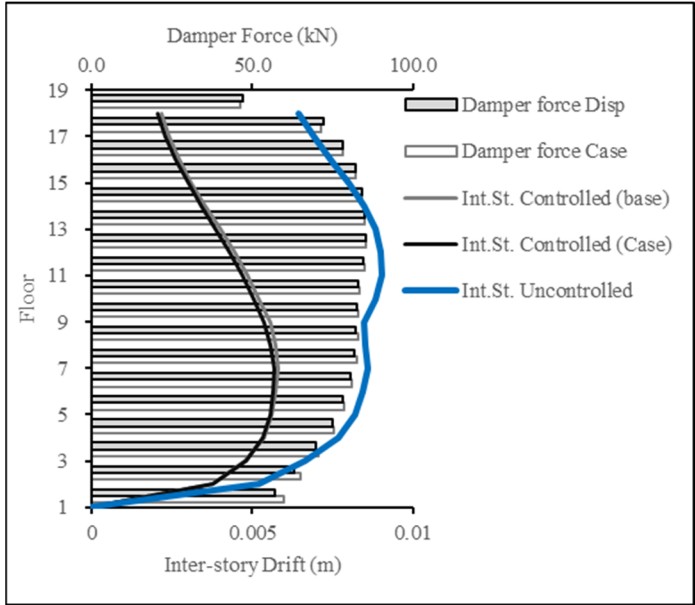

**Figure 27.** Inter-story drift reduction for ASCE EQ for 18-story building.

The rest of the figures present a selected collection of cases of choosing a single floor or multiple floors to guide the optimization algorithm towards reducing the displacement response for the chosen floor(s). From the presented results, it can be observed that clustering the target reduction for several consecutive floors presented more favorable results than scattering the selection or selecting a single floor for the response reduction. Selected cases are shown in Table 3.

**Table 3.** Selected floors case study.

| | Building 1 (9 Floors) | Building 2 (18 Floors) |
|---|---|---|
| Case | Target floors | Target floors |
| 1 | 3 | 6 |
| 2 | 6 | 12 |
| 3 | 9 | 18 |
| 4 | 3, 6 | 6, 12 |
| 5 | 3, 6, 9 | 12, 18 |
| 6 | 6, 9 | 6, 12, 18 |
| 7 | 3, 4 | 6, 7 |
| 8 | 3, 4, 5 | 6, 7, 8 |
| 9 | 6, 7 | 12, 13 |
| 10 | 6, 7, 8 | 12, 13, 14 |
| 11 | 2, 3, 4, 5 | 4, 5, 6, 7 |
| 12 | 6, 7, 8, 9 | 12, 13, 14, 15 |

Highlighted cases are presented below:
Highlighted cases for 9-story building are presented in Figures 28 and 29.

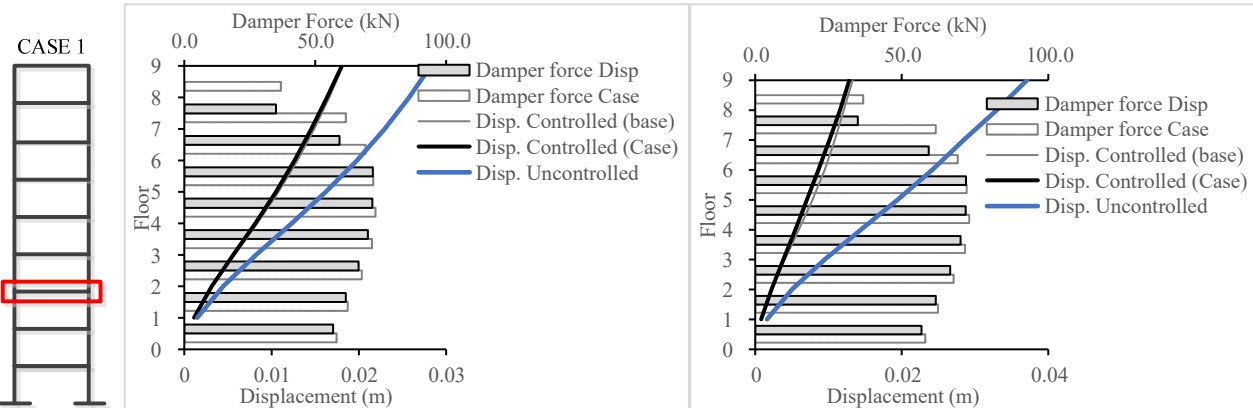

**Figure 28.** The simulation results for 1992 Egypt (**left**) and 1995 Aqaba (**right**) for 9-story building.

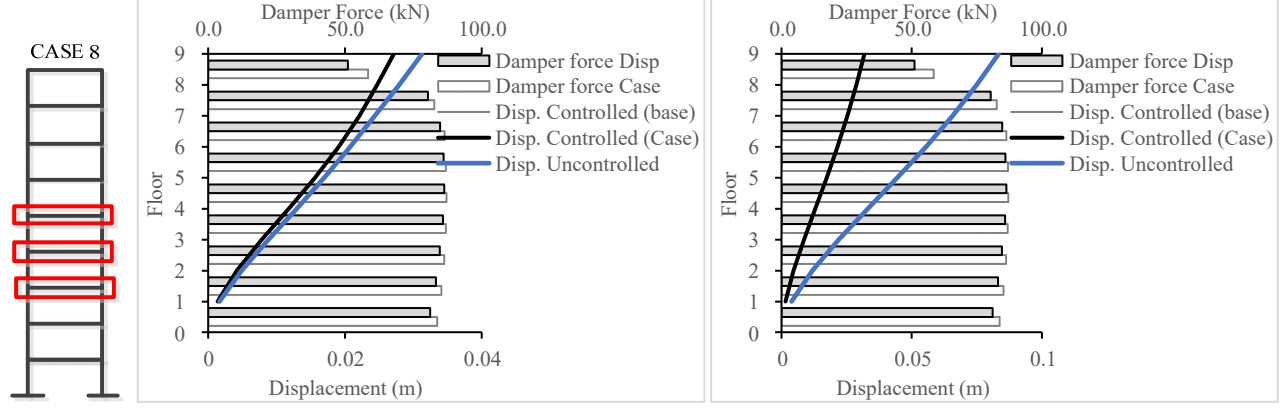

**Figure 29.** The simulation results for El Centro (**left**) and Northridge (**right**) for 9-story building.

The presented case in Figure 28 is the result for designing using the ECP EQ and the simulation using the 1992 Egypt earthquake and the 1995 Aqaba earthquake, respectively. For this case, an increase in the overall damper forces can be observed. The control scheme slightly enhanced the seismic response at level 3 but with much more damper force/effort. The displacement at level 3 decreased by 1.5%, while the overall displacement increased by 5.5%. This indicated a success in reduction for level 3 and presented an increase in other

levels, which were not specified for reduction. Similar results were obtained for the ASCE design step.

The presented case in Figure 29 is the result for designing using the ASCE EQ and the simulation using El Centro and Northridge, respectively. For this case, an increase in the overall damper forces can be observed. The control scheme slightly enhanced the seismic response at levels 3, 4, and 5 with slightly more damper force/effort. The displacement at levels 3, 4, and 5 decreased by 2.5%, 1.7%, and 1.2%, respectively, while the overall displacement increased by 1%. This indicated a success in reduction for levels 3, 4, and 5 and presented an overall increase in other levels, which were not specified for reduction. Targeting several floors presents better performance than targeting a single floor. Similar results were obtained for the ECP design.

Highlighted cases for 18-story building are presented in Figures 30 and 31.

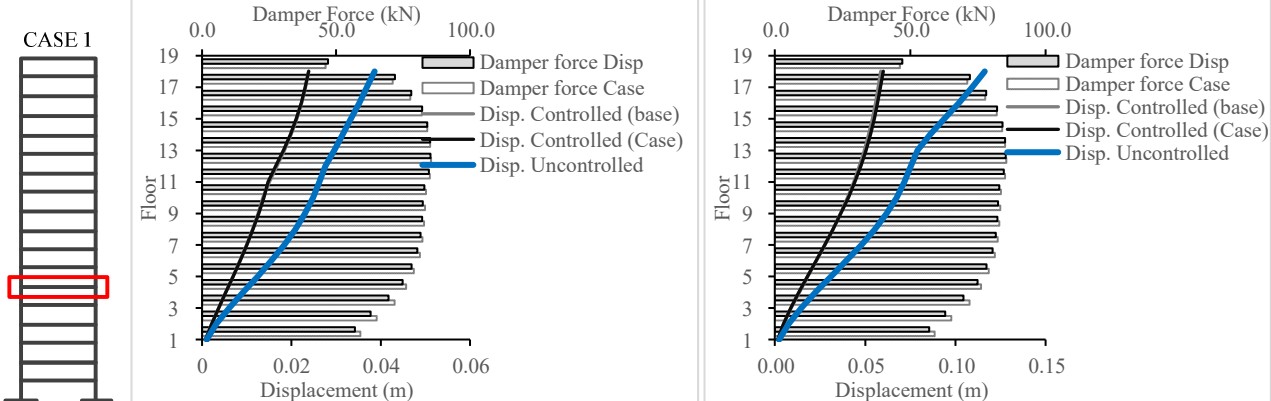

**Figure 30.** The simulation results for El Centro (**left**) and Northridge (**right**) for 18-story building.

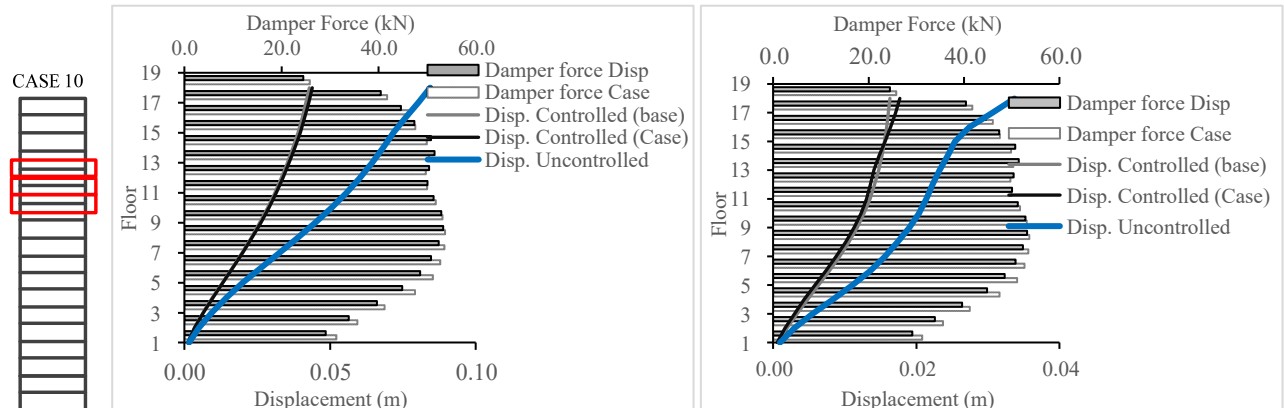

**Figure 31.** The simulation results for 1992 Egypt (**left**) and 1995 Aqaba (**right**) for 18-story building.

The presented case in Figure 30 is the result for designing using the ASCE EQ and the simulation using El Centro and Northridge, respectively. For this case, an increase in the damper forces can be observed up to level 6, above which damper forces decreased. The control scheme slightly enhanced the seismic response at level 6, with slightly more damper force/effort below level 6. The displacement at level 6 decreased by less than 1%, while the overall displacement increased by 1.6%. Similar results were obtained for the ECP design.

The presented case in Figure 31 is the result for designing using the ECP EQ and the simulation using the 1992 Egypt earthquake and the 1995 Aqaba earthquake, respectively. For this case, an increase in the overall damper forces can be observed. For the Aqaba EQ, the control scheme significantly enhanced the seismic response at levels 12, 13, and 14, but with more damper force/effort. The displacement at level 12 decreased by 5%, while the

overall displacement increased by 8%. This indicated a success in reduction for levels 12, 13, and 14 and presented an increase in displacement above level 14. This phenomenon was not observed for the 1992 Egypt EQ, where overall enhancement of the overall structural performance was observed with up to a 3% reduction in response when compared to the design phase.

## 7. Conclusions

The development of the GUI streamlined the design objectives guiding the process for MR-Damper placement in tall buildings. The LQG algorithm weighting matrices R and Q were constrained using a simplified interface that guided the PSO to produce favorable outcomes for specific floors in the structure. Based on the presented results, it can be concluded that the best practice for a Target behavior for a specific floor or set of floors is to cluster the Q bounding values in the GUI; this usually increases the forces in the MR-Dampers at the specific floor or set of floors and below. The target optimization J was adjusted to present three types of seismic response outcomes—acceleration, displacement, and inter-story drift—with success.

**Author Contributions:** Conceptualization, Y.F.R.; Methodology, A.F.F.; Validation, A.M.M.H.; Writing—original draft, A.A. All authors have read and agreed to the published version of the manuscript.

**Funding:** This project was supported financially by the Science and Technology and Innovation Funding Authority (STIFA), Egypt, Grant No. 37145.

**Data Availability Statement:** Not applicable.

**Conflicts of Interest:** The authors declare no conflict of interest.

## Appendix A. State Space Representation and LQG

The equation of motion of a multi degree of freedom system can be presented in a matrix format according to the Equation (A1) as presented:

$$M\ddot{U} + C\dot{U} + KU = \Gamma f - M\Lambda\ddot{x}_g \tag{A1}$$

where $M$ is the mass matrix, $C$ is the damping matrix, and $K$ is the stiffness matrix for the system. $\ddot{U}$, $\dot{U}$, and $U$ are the acceleration, velocity, and displacement vectors for the system degrees of freedom. The term $\Gamma$ is a vector that donates the locations and weights of an applied force vector $f$. The $\ddot{x}_g$ is the applied ground excitation, and vector $\Lambda$ is a vector representing the ground excitation directional force distribution.

The previous equation can be represented as a state-space formulation:

$x = \begin{bmatrix} U \\ \dot{U} \end{bmatrix}$ is called the state vector, and the change in state is presented as $\dot{x} = \begin{bmatrix} \dot{U} \\ \ddot{U} \end{bmatrix}$.

By rearranging 1, $\ddot{U}$ can be presented in the following format:

$$\ddot{U} = M^{-1}(-C\dot{U} - KU + \Gamma f - M\Lambda\ddot{x}_g) \tag{A2}$$

$$\ddot{U} = \begin{bmatrix} -KM^{-1} & -CM^{-1} \end{bmatrix} \begin{bmatrix} U \\ \dot{U} \end{bmatrix} + \Gamma M^{-1}f - \Lambda\ddot{x}_g \tag{A3}$$

Then state vector and system response $y$ are presented in the following equation:

$$\begin{aligned} \dot{x} &= Ax + Bf + E\ddot{x}_g \\ y &= Cx + Df + F\ddot{x}_g \end{aligned} \tag{A4}$$

where

$$A = \begin{bmatrix} 0 & I \\ -KM^{-1} & -CM^{-1} \end{bmatrix}, \quad B = \begin{bmatrix} 0 \\ \Gamma M^{-1} \end{bmatrix} \text{ and } E = \begin{bmatrix} 0 \\ -\Lambda \end{bmatrix}$$
$$C = \begin{bmatrix} -KM^{-1} & -CM^{-1} \end{bmatrix}, \quad D = \begin{bmatrix} \Gamma M^{-1} \end{bmatrix} \text{ and } F = \begin{bmatrix} -\Lambda \end{bmatrix} \tag{A5}$$

The control implemented applies a semi-active control strategy. The LQG is used. A combination of a Linear Quadratic Regulator (LQR) and a Linear Quadratic Estimator (LQE) using a Kalman filter for optimal state estimation are derived. The following cost function is to be minimized:

$$J = \lim_{t \to \infty} \frac{1}{t} \left[ \int_0^t \left\{ (Cx + Df)^T Q (Cx + Df) + f^T R f \right\} dt \right] \tag{A6}$$

where $R$ and $Q$ donate weighting matrices and the target of PSO ranges.

## Appendix B. Damper Model

The dynamic model used for the numerical analysis for the MR-Damper is the model proposed by Spencer. This phenomenological model is based on the Boc–Wen hysteresis model [48]. They presented a prototype MR-Damper with a maximum produced force equal to 1000 N. The dynamic factors effecting the behavior of the damper are presented in Figure A1, where the Bouc–Wen model, the MR fluid stiffness $k_o$, and the damping $c_o$ are between displacement $x$ and $y$, while the additional dashpot $c_1$ and spring $k_1$ represent the force-roll-off behavior at low velocities and the accumulator unit stiffness, respectively. Force in this damper is obtained through a set of equations presented by Spencer:

$$F = c_1 \dot{y} + k_1 (x - x_o) \tag{A7}$$

Consider the equivalent forces at the upper part of the damper:

$$c_1 \dot{y} = \alpha z + k_o(x - y) + c_o(\dot{x} - \dot{y}) \tag{A8}$$

where z is the evolutionary parameter governed by

$$\dot{z} = -\gamma |\dot{x}|z|z|^{n-1} - \beta \dot{x}|z|^n + A\dot{x} \tag{A9}$$

and $\alpha$ is the pre-yield to post-yield ratio. The values $n$, $\gamma$, $\beta$, and $A$ are the parameters that control the hysteresis behavior, and they are obtained through experimental means [48].

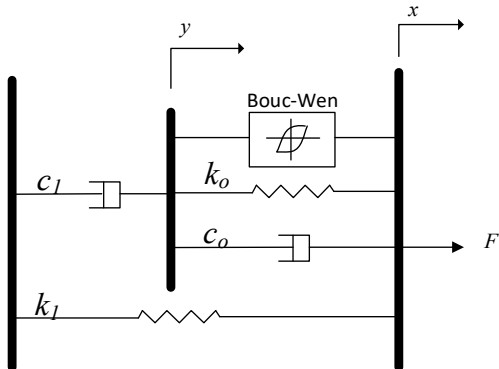

**Figure A1.** Modified Bouc–Wen model proposed by Spencer [48].

Large MR-Damper-Modified Bouc–Wen parameters were presented by Jung. They are of 100 kN capacity [49] and were obtained by scaling a 20 kN MR-Damper 5 times for the force and 2.5 times for the stroke of the device. The parameters for the MR-Damper used in this study are presented in Table A1.

**Table A1.** The large MR-Damper parameters obtained from Jung [49].

| Parameter | Value |
|---|---|
| $\alpha_a$ | 46.2 kN/m |
| $\alpha_b$ | 41.2 kN/m/V |
| $c_{0a}$ | 110.0 kN s/m |
| $c_{0b}$ | 114.3 kN s/m/V |
| $c_{1a}$ | 8359.2 kN s/m |
| $c_{1b}$ | 7482.9 kN s/m/V |
| $\eta$ | 100 s$^{-1}$ |
| $k0$ | 0.002 kN/m |
| $k1$ | 0.0097 kN/m |
| $\gamma$ | 164.0 m$^{-2}$ |
| $\beta$ | 164.0 m$^{-2}$ |
| $A$ | 1107.2 |
| $n$ | 2 |

In this model, three parameters depend on the current driver $u$:

$$\alpha(u) = \alpha_a + \alpha_b u, \; c_o(u) = c_{oa} + c_{ob}u, \text{ and } c_1(u) = c_{1a} + c_{1b}u \tag{A10}$$

The first order filter is $\dot{u} = -\eta(u - v)$, where $v$ is the applied voltage to the current driver that donates the current involvement in the MR fluid to reach rheological equilibrium [48].

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
