# Peer review of "Adjustment of Tall Building Behavior by Guided Optimization of Magneto-Rheological Damper Control Parameters"

_2673-4109, doi:10.3390/civileng4020035_

Round 1

Reviewer 1 Report

Adjustment of Tall Buildings Behavior by Guided Optimiza- 2 tion of Magnetorheological-Dampers Control Parameters

Authors:  Amin Akhnoukh, Ahmed Fady Farid, Ahmed M.M. Hassannand Youssef F. Rashed

The paper presents a a scheme based on Magneto-rheological dampers to reduce the dynamic response on buildings. The focus is on the design methodology for the controller. The optimization seeks to change the dynamic behavior for specific floors of the building. 17 parameters are included as design variables.

Review.

In general the paper is well written in its present force

The authors present a methodology mostly favorable to an easy computational implementation. There is no real innovation in terms of theory or control lawsa

Design phase is the most interesting part. There I have observations.

·       Why did the authors choose genetic algorithms and particle swarm algorithms. Is the problem with local minima too critical to use first order optimization?

·       How much computation effort is involved in this phase?

·       The authors use LQG in order to determine the control law adjusting the weight matrices to obtain acceptable designs. LQG is not very good at handling transient response The authors should comment on that.

·       Which not consider the input directly into the control law instead of LQR

The best part of the paper is the case study. Well analyzed

Author Response

  1. The authors thank the reviewers for their comments. It has to be noted that the presented work is a continuation of previously published work. However, the structural dynamic control scheme is a stochastic problem, since the driving force i.e., EQ is also a stochastic problem. Without a predefined gradient to the response results, damper placement, and optimal control parameters, population algorithms such as (GA/Swarms) were found to best suit our scheme.
  2. The computation effort for this stage depends on several factors including:

    1. the size of the EQ in terms of number of record points.
    2. the size of the structure (Stiffness, Mass, and Damping matrices).
    3. the number of required Swarms particles, number of iterations.
    4. the number of required iterations for the GA.

    In terms of time consumption, one swarms iteration for a 1000-point EQ for the 9 story building is done in slightly under 1 min. The process relies heavily on the performance of the GA part.

  3. The authors thank the reviewers for their comments. LQG (Linear Quadrature Gaussian) is best suited for steady state systems and is not very good at handling transient response. However, the problem at hand consists of two parts, the first part is the structure while subjected to ground excitation (forced vibration mode/transient response mode). The second part is the structure continuous vibration after the end of ground excitation (free vibration mode/steady state mode). The controller should be able to handle both modes. LQG was found to be suitable for both modes. Both modes are equally significant in seismic design.

Reviewer 2 Report

The paper "Adjustment of Tall Buildings Behavior by Guided Optimization of Magnetorheological-Dampers Control Parameters" is interesting but requires more attention to the presentation of the subject and some observations are presented below:

- What is the connection "MR - damper structure" from Figure 2 with the studied case, briefly presented in Figures 8-9. Are these shock absorbers (from Figure 2) inserted somewhere in the structure of the simulated buildings (Figures 8-9)?

- It is not understood how figures 10 and 11 resulted;

- The bibliographic references (sources) of figures 12, 13, 14, 15 are missing;

- in figure 19, the blue curve is missing from the legend;

- in figure 26, the blue curve is missing from the legend;

- After Figure 26, another figure appears, unnumbered, which also lacks the blue curve from the legend;

- From the following figures, which occupy 4 pages (pages 17-20), the authors should choose a maximum of 4 representative figures, which must be numbered and in which the obtained results should be explained.

- For most of the graphical results obtained, there are no comments regarding the obtained results, possibly a comparison with other similar results from the specialized bibliography.

- The conclusions part is insufficiently developed and the contributions of the authors are not highlighted in correlation with the results presented in this paper.

Author Response

  1. The MR-damper structure presented in Figure 2 is an example for the MR-dampers that are inserted in the structures that are analyzed in this paper (structures typical floor plan is shown in Figure 8 and the building elevation is shown in Figure 9).
  2. Figures 10 and 11 are generated by reporting the acceleration values in meter/sec2  (on y-axis) for the response of the building (in Figures 8 and 9) when the building was exposed to earthquake (with parameters generated using the Egyptian Code of Practice for Figure 10) and the ASCE parameters (for Figure 11).
  3. The sources (references) are added to the text.
  4. In Figure 19, the blue curve is added to the legend.
  5. Im Figure 26, the blue curve is added to the legend.
  6. Figures on page 17-20 are curtailed to 4 figures per building case (9 stories or 18-stories). Selected figures/cases are shown in a newely generated table (Table #3). Sufficient comments are made on the figures which clarify the research findings (conclusions).

Round 2

Reviewer 2 Report

After analyzing this paper, I found that the authors made some requested changes.

I believe that the paper can be accepted for publication in this form.